# Multi-strategy evolutionary games: A Markov chain approach

**Mahdi Hajihashemi**📷*, **Keivan Aghababaei Samani**📷

Department of Physics, Isfahan University of Technology, Isfahan, Iran

* mehdi.hajihashemi@alumni.iut.ac.ir

## Abstract

Interacting strategies in evolutionary games is studied analytically in a well-mixed population using a Markov chain method. By establishing a correspondence between an evolutionary game and Markov chain dynamics, we show that results obtained from the fundamental matrix method in Markov chain dynamics are equivalent to corresponding ones in the evolutionary game. In the conventional fundamental matrix method, quantities like fixation probability and fixation time are calculable. Using a theorem in the fundamental matrix method, conditional fixation time in the absorbing Markov chain is calculable. Also, in the ergodic Markov chain, the stationary probability distribution that describes the Markov chain's stationary state is calculable analytically. Finally, the Rock, scissor, paper evolutionary game are evaluated as an example, and the results of the analytical method and simulations are compared. Using this analytical method saves time and computational facility compared to prevalent simulation methods.

**Data Availability Statement:** Codes related to this article is hosted on Github at https://github.com/mehdiphy/rock-scissors-paper-evolutionary-game.

## Introduction

Today, Evolutionary Game theory (EGT) is a progressive topic in many branches of science from economy to biology [1–10]. EGT provides powerful tools for many problems in which the system's dynamics depend on the interaction between agents. The interactions between strategies are often described by evolutionary games. The performance of strategies in evolutionary games is determined by the game's payoff matrix, which determines each strategy's spread rate. Greater payoff in the game leads to more tendency to spread in the population for any strategy. In an infinite well-mixed population, dynamics of the system is governed by a deterministic equation called replicator equation [11, 12], but in a finite population the dynamics is stochastic [13–21].

In a stochastic evolutionary game, the population is divided into several strategies and individuals interact with each other based on their strategies. The process is advanced by discrete time steps. In each time step, the frequency of strategies changes by one or remains unchanged. The game's payoff matrix and frequency of each strategy identify the probability of events at each time step. Another factor that influences the dynamics of the population is the update rule. Update rule identifies how payoff matrix and frequencies distribute the probabilities of events in each time step. Depending on the update rule, the evolutionary game can be stopped when one of the strategies overcomes all other strategies (fixation), or continues forever. The

**Funding:** This research is financially supported by Iran National Science Foundation (INSF) under Postdoctoral Research Grant number 99007738.

**Competing interests:** The authors have declared that no competing interests exist.

structure of the population can also affect the dynamics of population. Unfolding an evolutionary game in graph-structured populations is the subject of many investigation [13, 22–29]. Cooperative behaviors in games like public good game or prisoner's dilemma is a charming topic in evolutionary games investitations [30–36].

In stochastic evolutionary games, the fixation of a strategy is the favorite subject. Numerical simulation is the subject of many studies in finite populations [37–40]; also there are many investigations that evaluated the dynamics of evolutionary games analytically [18, 41–46]. In analytical evaluation, the evolutionary process is often considered as a generalization of the Moran process [47], and it has been done for games with two strategies. The most famous analytical method for analyzing evolutionary games is the recursive equation method [48, 49]. In this method, two interesting quantities, fixation probability and fixation time obtain in terms of finite series. Evolutionary games with more than two strategies are not studied analytically so far.

Considering the individual's mutation, the population's dynamics are governed by an evolutionary game with no fixation strategy. So, after many time steps, the configuration of the population reaches a stable state. This steady state is described by a stationary probability distribution which determines after a long run, each configuration of population how much is possible. In both cases (games with fixed strategies and games with no fixed strategies), as the number of strategies increases, more time and computational facilities are needed for simulation of the evolutionary game, so proposing an analytical method for evaluating evolutionary games with more than two strategies is helpful. This study aims to provide an analytical method for obtaining concepts in evolutionary games that getting them by simulation takes long time and needs extensive computational facilities.

Markov chain method has been used for analyzing evolutionary games sincessfully [50–52] but it has never been used in an organized and intensive way. In this paper we stabilize the Markov chain method as a reliable method for evaluating evolutionary games. In this method corresponding to each evolutionary game, a Markov chain is introduced. Essential concepts in evolutionary games such as fixation probability, conditional fixation time, and stationary probability distribution are related to concepts in the Markov chain. Using the fundamental matrix method in the equivalence Markov chain, we can calculate essential concepts in the Markov chain, which leads to calculating essential quantities in the evolutionary game. Although this method is designed for a discrete-time system, it could be used for a time-continuous system by considering some approximation.

The organization of the paper is as follows. In general method section we review the Markov chain method and explain a practical theorem for obtaining conditional fixation times which is proven in tha Appendix. In evolutionary game section we establish correspondence between evolutionary games and Markov chains and will clarify how essential concepts in evolutionary games can be obtained from the fundamental matrix method. In result and discussion we apply our approach to an evolutionary game with three strategies. Here the famous rock, scissor, paper evolutionary game is used and results of analytical method and simulations are compared to each other. Conclusion is devoted to a summary and concluding remarks.

## General method

### Markov chain and fundamental matrix method

In this section, We briefly review the fundamental matrix method in Markov chains and obtain a formula for calculating conditional absorption time. In the next section, by establishing a correspondence between states of the Markov chain and states of the evolutionary game, this theorem provides handy information about the dynamic of the evolutionary process in the fixation path.

A Markov chain is described as $S$ set of states $S = \{s_1, s_2, s_3, \ldots\}$ and a process which starts in one of these states and move to another state. If the chain is currently in state $s_i$, then it moves to state $s_j$ with probability denote by $p_{ij}$. The point is that the probability that the chain moves from state $s_i$ to state $s_j$ depends on the initial state $s_i$ and final state $s_j$ not upon which states the chain was in before the state $s_i$. The probabilities $p_{ij}$ constructed the transition matrix $P$. If $v_i$ be a vector that determines the probability distribution in step $i$, then probability distribution in step $i + 1$ is $v_{i+1} = v_i P$. If there are states in the Markov chain that leaving these states is impossible, these states are called absorbing states and Markov chain called absorbing Markov chain. If $i$ be an absorbing state, then $p_{ii} = 1$ and when the chain is in this state, the Markov chain ends. Other states which are not absorbing are called transient. There are three valuable concepts related to absorbing Markov chain. The first is the probability that the chain starts from transient state $i$, will be absorbed in absorbing state $j$ ($b_{ij}$). The second is absorption time ($t_i$), the expected number of steps before the chain is absorbed in one of absorbing states, given that the chain starts from state $i$ and the last concept is conditional absorption time ($\tau_{ij}$), the expected number of steps before the chain is absorbed in the absorbing state $s_j$ given that the chain starts in transient state $i$. It is necessary to emphasize that absorption time differs from conditional absorption time. In fact, absorption time is a weighted average of conditional absorption time among different absorbing states. There is a helpful method for calculating absorption probabilities and absorption time called the fundamental matrix method. In this method, at first the transition matrix is written in the canonical form as follows:

$$P = \left( \begin{array}{c|c} Q & R \\ \hline 0 & I \end{array} \right).$$

(1)

In other words, in canonical form, we labeled states so that the absorbing states consider as final states. The so-called fundamental matrix is defined as $N = (I - Q)^{-1}$ and is useful to obtain absorption probabilities and absorption time. Let us define $t_i$ to be the (average) absorption time of the Markov chain starting from state $i$ and $\rho_i^{a_1}, \rho_i^{a_2}, \ldots$ the absorption probabilities correspond to absorption states $a_1, a_2, \ldots$ starting from state $i$, respectively. According to the approach in Ref [53] the matrix notation can be used to denote these quantities:

$$t = \begin{bmatrix} t_1 \\ t_2 \\ \vdots \\ t_T \end{bmatrix}, B = \begin{bmatrix} \rho_1^{a_1} & \rho_1^{a_2} & \cdots \\ \rho_2^{a_1} & \rho_2^{a_2} & \cdots \\ \vdots & \vdots & \\ \rho_T^{a_1} & \rho_T^{a_2} & \cdots \end{bmatrix}.$$

where in the above $T$ is the number of transient states. Using the fundamental matrix method one can obtain absorption probabilities and times as follow:

$$B = NR,$$
$$t = Nc.$$

(2)

where $c = (1, 1, \cdots, 1)^t$. If there is no absorbing state in Markov chain then the Markov chain is called ergodic. In the ergodic Markov chain, it is possible to go from every state to every other state after finite steps. If $P$ is the transition matrix of the ergodic Markov chain then for $n \to \infty$ the $P^n$ approach a limiting matrix $W$ with all rows the same vector $w$, called fixed row vector for $P$. It means after a long run, the Markov chain reaches an equilibrium which

probability that chain be in state $j$ determine by $w_j$. Obviously, $wP = w$ means $w$ is the left null vector of matrix $P - I$.

$$w(P - I) = 0 \tag{3}$$

In other words, the fixed row vector of $P$ is left eigenvector of $P$ with eigenvalue one. The fundamental matrix method does not represent a recipe for calculating the conditional fixation time. Now we describe a theorem to calculate the conditional fixation time for any absorbing state by adding some details to the fundamental matrix method.

Theorem: Let $\tau_{ia}$ be the conditional fixation time for absorption in absorbing state $a$ given that Markov chain starts from transient state $i$. Using matrix notation, we have

$$\tau_a = N^a c. \tag{4}$$

where in above equation

$$\tau_a = \begin{bmatrix} \tau_{1a} \\ \tau_{2a} \\ \vdots \\ \tau_{Ta} \end{bmatrix}, c = \begin{bmatrix} 1 \\ 1 \\ \vdots \\ 1 \end{bmatrix}, N_{ij}^a = \frac{\rho_j^a}{\rho_i^a} N_{ij}$$

and $T$ is the number of transition states.

The proof of this theorem present in the Appendix. Also there is a proof with different notation for the theorem in Ref [54].

## Evolutionary games corresponding Marokov chains

This section develops a method based on correspondence between Markov chain dynamics and evolutionary game dynamics. This correspondence provides a sound mathematical device for analyzing evolutionary games.

Consider a population with size $N$ which $n$ strategies interact with each other according to a payoff matrix

|        | $S_1$    | $S_2$ | . | . | . | $S_n$    |
|--------|----------|-------|---|---|---|----------|
| $S_1$  | $a_{11}$ | . | . | . | . | $a_{1n}$ |
| $S_2$  | $a_{21}$ | . | . | . | . | $a_{2n}$ |
| .      | . | . | . | . | . | . |
| .      | . | . | . | . | . | . |
| .      | . | . | . | . | . | . |
| $S_n$  | $a_{n1}$ | . | . | . | . | $a_{nn}$ |

In each time step, the expected payoff of each strategy is obtained in terms of frequency of strategies and payoff matrix as

$$\pi(i) = \frac{1}{N-1} \sum_{j=1}^{n} a_{ij} f_j \qquad (5)$$

where $\pi(i)$ is excepted payoff of strategy $i$ and $f_j$ is frequency of strategy $j$. Generally, the expected payoff interpreted as the fitness of strategy in evolutionary game theory, in other words, strategies spread with rates that are proportional to their expected payoff. There are many ways to obtain the fitness of a strategy from its expected payoff, like an exponential payoff to fitness mapping. Depending on the update rule of dynamic, there is a possibility that the evolutionary process leads to the fixation of a strategy which means one strategy overcomes other strategies and occupies the whole population forever. In evolutionary games with fixation strategies, three concepts are noteworthy. Fixation probability, the probability that a strategy fix in population, the fixation time, the average steps of time that an evolutionary process fixed to one of its fixation strategies and conditional fixation time, the average steps of time that evolutionary game fixed in a specific strategy. Update rule could be in such a way that there is no possibility for any strategy that overcomes other strategies forever. In this situation, after a long run with many steps of time, the population reaches to stable condition, which means the probability that the evolutionary process is in each state approach a stationary value.

In the evolutionary process the state of population describe by frequency of each strategy like $\{f_{s1}, f_{s2}, f_{s3}, \ldots\}$ which $f_{s1} + f_{s2} + \ldots f_{sn} = N$. Direct calculation shows that the number of states is

$$\binom{N+n-1}{n-1}. \qquad (6)$$

In each time step, one strategy is chosen for reproduction and replaces its offspring inplace another strategy. In other words, in each step, the frequency of a strategy increases by one, and frequency of another strategy decreases by one, and the state of the evolutionary game changes. Update rule of the evolutionary game determines which strategy has a higher probability of reproduction and which strategy has a higher probability of being replaced. It is possible that the strategy that is chosen for reproduction and the strategy that vanishes be the same, in this situation, the state of the evolutionary game remain unchanged.

Corresponding to each evolutionary game with $l$ fixation strategies, there is a Markov chain with $l$ absorption states, also, corresponding to each evolutionary game with no fixation strategy, there is an ergodic Markov chain. States in evolutionary game dynamic can be considered as Markov chain states. Transition matrix of corresponding Markov chain obtains by update rule of the evolutionary game.

Fixation probability, Fixation time, and conditional fixation time in the evolutionary game correspond to absorption probability, absorption time, and conditional absorption time in the Markov chain. Since we have the fundamental method in Markov chain theory, this duality between Markov chain dynamics and evolutionary game dynamics is so helpful to analyze the evolutionary games. In games with fixation strategies using the theorem of section, one can obtain conditional fixation time for each strategy, and in evolutionary games with no fixation strategy, the stationary probability distribution of strategies is obtained by calculating the left null vector of matrix $P - I$. In the next section, we use this correspondence for analyzing rock, scissor paper game.

## Results and discussion

As an example of what we said, in this section, we analyze the most famous game with three strategies, the rock, scissors, paper game (RSP game) [4, 55–67]. In the RSP game, each strategy overcomes the next one cyclically.

In the real world, coexistence of many species occurs over three competing species interacting with each other like the rock-paper-scissors game. According to the anticipation of some models, the coexistence of all three competitors is possible if the interaction between them becomes local. In reference [68], the coexistence of three populations of Escherichia coli was empirically studied. According to this, coexistence is preserved when the interaction between species is localized. When dispersal and interaction are nonlocal, the diversity is lost, and one species occupies the whole population. Another example of the rock-paper-scissors evolutionary game in biology is changing in the frequency of adult side-blotched lizards. In reference [69], the authors studied the frequencies of three side-blotched lizard morph from 1990-95. According to their observations, the fitness of each morph is dependent on other morphs. They suggest an evolutionary stable strategy model which predicts each morph frequency. Estimating parameters of payoff matrix of RSP game by field data, the model predicted the morphs oscillation frequencies.

Without loss of generality, the payoff matrix of RSP game can be depicted as follow

|   | P | S | R |
|---|---|---|---|
| P | 0 | $-a_2$ | $b_3$ |
| S | $b_1$ | 0 | $-a_3$ |
| R | $-a_1$ | $b_2$ | 0 |

where $a_i$, $b_i > 0$. At first, we set the update rule so that the evolutionary process ends when the whole population occupies by one strategy. Therefore, the Markov chain is an absorbing Markov chain. By changing the update rule of the evolutionary game, we establish the possibility of mutation, which means when a strategy extincts, there is a probability that other strategies mutate to extincted strategy and it appears in the population again. In this situation, the evolutionary process never ends but after a long run, it reaches a stable position, and the corresponding Markov chain is an ergodic Markov chain.

### RSP game with absorbing states

Consider a population with size $N$ that each member of the population can be one of three types rock, scissor, and paper. We denote the three strategies rock, scissors, and paper as 1, 2 and 3, respectively. The evolutionary process runs upon a birth and death update rule. According to this update rule, one member of the population is chosen for reproduction at each step of time. The chosen member selects randomly another member of the population to be replaced with its offspring. The probabilities of selection for reproduction and being replaced for each strategy are proportional to their frequency. The expected payoff of each strategy is involved in the update rule via the Fermi distribution function [70]. The probability that in each step of time, strategy $k$ replaced with strategy $l$ is

$$p_{k \longrightarrow l} = \frac{2f_k f_l}{N(N-1)} \times F(\pi_k, \pi_l) \tag{7}$$

which $f_k$ and $f_l$ are frequency of strategies $k$ and $l$ respectively and $F$ is fermi function define as

follow

$$F(x,y) = \frac{1}{1 + e^{\beta(x-y)}} \tag{8}$$

where $\beta > 0$ is constant. The expected payoff for strategies ($\pi_k$) can be calculate for $k = 1, 2, 3$ as

$$
\begin{aligned}
\pi_1(f_1, f_2, f_3) &= \frac{b_3(N - f_1 - f_2) - a_2 f_2}{N} \quad , \\
\pi_2(f_1, f_2, f_3) &= \frac{b_1(N - f_1) - a_3(N - f_1 - f_2)}{N}, \\
\pi_3(f_1, f_2, f_3) &= \frac{b_2 f_2 - a_1 f_1}{N}.
\end{aligned}
\tag{9}
$$

According to Eq (7), when a strategy extincted, there is no possibility that appears again in the population, and sooner or later, the whole population occupies by one of the strategies. It means the corresponding Markov chain is an absorption Markov chain. According to Eq (6) the number of states in this Markov chain is $\frac{(N+1)(N+2)}{2}$. States of the Markov chain can be arranged in an equilateral triangle. Fig 1 shows the corresponding Markov chain of the evolutionary game with this specific update rule for $N = 10$. Arrows show the allowed transitions between states. Fig 1 can also be considered as a simplex that determined states of the evolutionary game. The vertices of the triangle are absorption states that are correspond to fixation

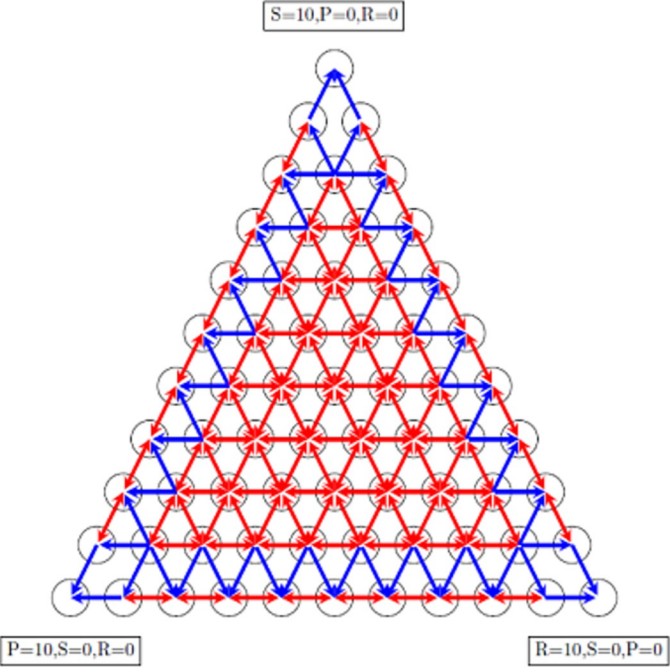

**Fig 1. The absorbing Markov chain corresponds to an evolutionary RSP game with a population N = 10.** The total number of states is 55, and the arrows determine the allowed transitions between states. Some arrows are two-way, and some of them are one-way arrows. Inside the simplex, all the states are transient, and transitions between them are two ways. Transitions between the inside of simplex and sides are one way. It means when the Markov chain is in the states of sides, it never goes back inside the simplex. In other words, when a strategy extinct, it never appears in population anymore. Transitions between states of sides are two-way too, except transitions between absorption states and their neighbors, which is one way.

strategies in the evolutionary game. When Markov chain is on the triangle's sides, it is impossible to return inside the triangle because by this specific update rule, when a strategy extinct, it never comes back. When Markov chain is on a triangle's side, it absorbs in one of two vertices side. We are interested in obtaining fixation probability and conditional fixation time for any state in the simplex. After constructing the transition matrix using Eq (7) and calculating the fundamental matrix, one can obtain the fixation probability of every state of simplex for three absorption states.

After finding the fixation probability of states, by using the theorem of section, we can obtain conditional fixation time for any state of the simplex.

To observe the footprint of the RSP game, we set the elements of the payoff matrix in the neutral case and strong selection both. In the neutral case, the elements of payoff matrix are $a_1 = a_2 = a_3 = 1$, $b_1 = b_2 = b_3 = 2$. In the strong selection case, we set the elements of the payoff matrix extremely in favor of the paper strategy and in detriment of rock strategy. In this case, we have $a_1 = a_3 = 1$, $a_2 = 300$, $b_1 = b_3 = 0$, $b_2 = 300$. Figs 2–4, show the fixation probability of paper, scissors and rock strategies respectively, when the process begins in each state in the simplex. In the neutral selection case, when the distance between the beginning state and absorption state decreases, the probability of absorption increases. After changing the payoff matrix in favor of the strategy paper, the probability of absorption to the strategy paper increased for all states inside the simplex. In this case, states with long distance to fixation state $R = 0$, $S = 0$, $P = N$ also have a high probability of absorbing to this fixation state.

Also there are states that have high probability of absorbing to scissors strategy in neutral case, but in strong selection case, they have high probability of absorbing to paper strategy. That is because the payoff matrix changed in favor of paper strategy. Also, some states have a high probability of absorbing to rock strategy in neutral case, but in the strong selection case, they have a high probability of absorbing to scissors strategy. That is because we changed the payoff matrix to the detriment of the rock strategy. In the strong selection case, there are fewer states with a high probability of absorbing to rock strategy. Changing the payoff matrix has effects on conditional fixation time too. Figs 5–7, show the conditional fixation time of paper, scissors and rock strategy respectively, when the process begins in each state in the simplex.

Comparing conditional fixation time in the neutral and strong selection cases shows that absorption to the paper strategy happens in a shorter time in the strong selection case. As

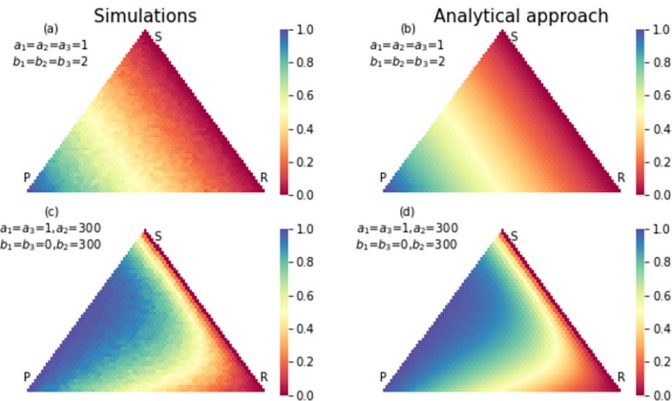

**Fig 2.** The simulation (a) and analytical (b) results for fixation probability of strategy paper in an RSP game with neutral selection. The size of population is 50. The states close to absorption state $P = N$, $S = 0$, $R = 0$ have a higher chance of absorbing in this absorption state. In (c) and (d) same results were shown with a strong selection in favor of paper strategy and detriment of rock strategy. Compared to the neutral case, many states have a higher chance to absorb to paper strategy.

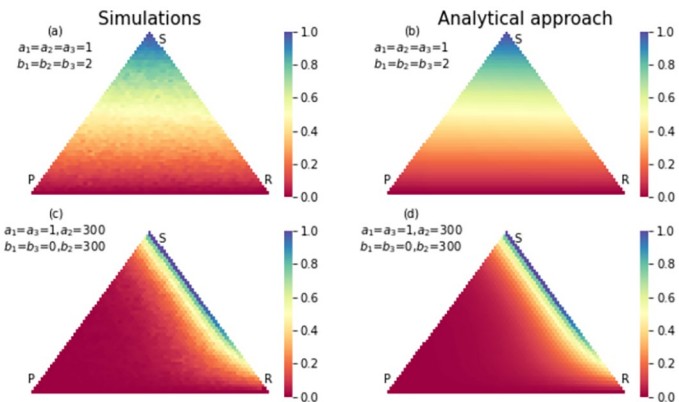

**Fig 3.** The simulation (a) and analytical (b) results for conditional fixation time of strategy paper for an RSP game with a neutral selection. The size of the population is 50. No wonder that states are close to absorption state $P = N$, $S = 0$, $R = 0$ reaches this absorption state by the fewer steps. In (c) and (d) same results were shown for a strong selection in favor of paper strategy and detriment of rock strategy. Compared to the neutral case, the number of steps for reaching $P = N$, $S = 0$, $R = 0$ is reduced due to strong selection in favor of strategy P.

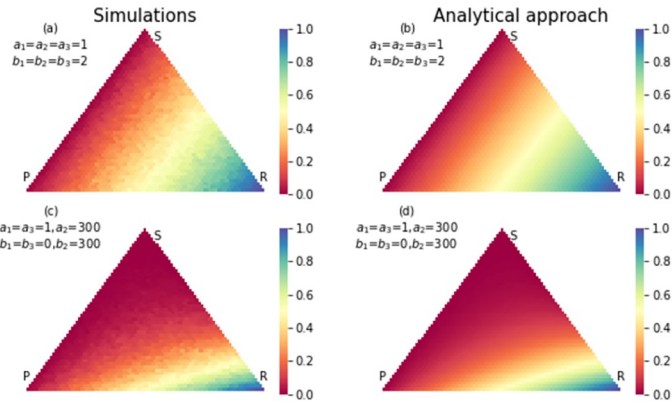

**Fig 4.** The simulation (a) and analytical (b) results for fixation probability of strategy rock in an RSP game with neutral selection. The size of the population is 50. In (c) and (d) same results were shown with a strong selection in favor of paper strategy and detriment of rock strategy. Since the payoff matrix is in detriment of rock strategy, many states, even those who are close to absorption state $P = 0$, $S = 0$, $R = N$ have fewer chances to absorb to $P = 0$, $S = 0$, $R = N$.

shown in Fig 6, the states which are close to fixation state $R = N$, $S = 0$, $P = 0$, in the strong selection case, absorb in strategy scissors in a shorter time. Also, the conditional fixation time for absorbing in the rock strategy increases in the strong selection case for all simplex states. The reason again is changing the payoff matrix to the detriment of rock strategy. In all figures, the results from the analytical approach and simulations are compared to each other. In most of them, simulation results coincide with analytical results. Still, in Figs 6 and 7 in the part of strong selection, the similarity is not so obvious. Since in some states, the probability of absorption to rock strategy is very low in the strong selection case, we need a lot of realization of the evolutionary game to reach a limited number realization ended in rock strategy. It means simulation should repeat more times for obtaining an accurate result. The same is true for conditional fixation time of scissors strategy. The hardness of getting simulation results in some conditions shows the necessity of invent of an analytical method.

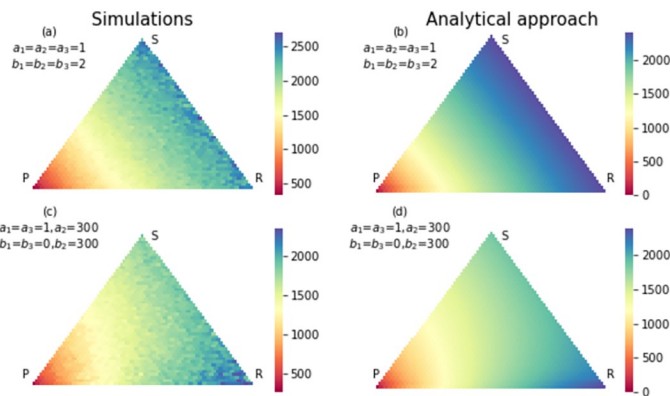

**Fig 5.** The simulation (a) and analytical (b) results for fixation probability of strategy paper in an RSP game with neutral selection. The size of population is 50. The states close to absorption state $P = N, S = 0, R = 0$ have a higher chance of absorbing in this absorption state. In (c) and (d) same results were shown with a strong selection in favor of paper strategy and detriment of rock strategy. Compared to the neutral case, many states have a higher chance to absorb to paper strategy.

## RSP game without absorbtion states

One may set the update rule in such a way that none of the strategies fix forever. In this situation, the corresponding Markov chain is an ergodic Markov chain. To compare our final result to the numerical result obtained in previous works, we use the update rule of Ref. [39]. According to this update rule, the probability that in each time step, one member of the population switches from strategy $l$ to strategy $k$ is proportional to $T_{l \to k} = \varepsilon + W(\pi_k - \pi_l)$ where $\varepsilon$ is a positive value which guarantees mutation in the process and $W$ is zero when the argument is negative. $W$ works like the identical function when the argument is positive or zero. Elements of the transition matrix can be calculated as follow

$$p_{l \to k} = \frac{T_{l \to k}}{\sum_{ij} T_{i \to j}} \tag{10}$$

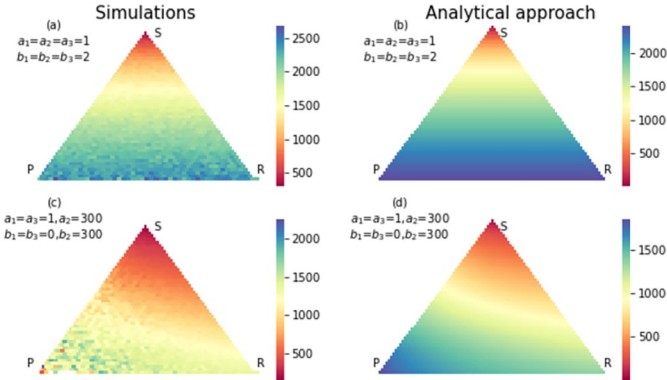

**Fig 6.** The simulation (a) and analytical (b) results for fixation probability of strategy scissors in an RSP game with neutral selection. The size of the population is 50. The states close to absorption state $P = 0, S = N, R = 0$ have a higher chance of absorbing in this absorption state. In (c) and (d) same results were shown with a strong selection in favor of paper strategy and detriment of rock strategy. Compared to the neutral case, some states are close to $P = 0, S = 0, R = N$ but have a high chance to absorb in $P = 0, S = N, R = 0$ strategy. The reason is imposing strong selection to the detriment of rock strategy.

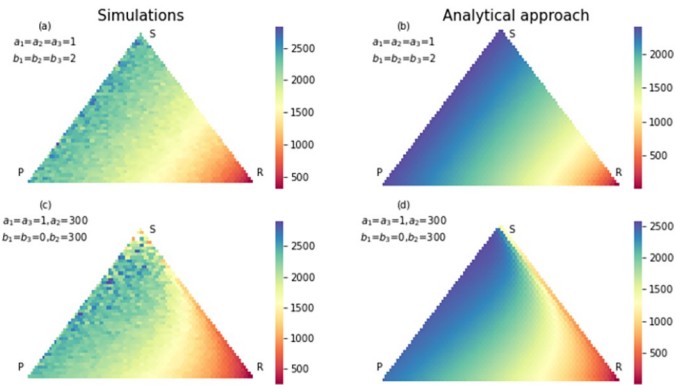

**Fig 7.** The simulation (a) and analytical (b) results for conditional fixation time of strategy rock for an RSP game with a neutral selection. The size of the population is 50. In (c) and (d) same results were shown for a strong selection in favor of paper strategy and detriment of rock strategy. Due to strong selection against strategy rock, conditional fixation time increase for all states of simplex.

where $i \rightarrow j$ are all allowed transitions in each state of the Markov chain. Fig 8 shows the Markov chain corresponding to this update rule. Unlike the previous update rule, when the Markov chain is in any state, there is a nonzero probability that exits from that state and therefore, there is no absorption state. The limiting probability distribution of the evolutionary game can be obtained by calculating the left null vector of matrix $P - I$. Fig 9 shows the analytical and simulation results for limiting probability distribution after a long run (100 million steps). The simulation and analytical results agree with each other. As a double-check, one can compare the results with simulation results obtained with the same update rule in Re. [39].

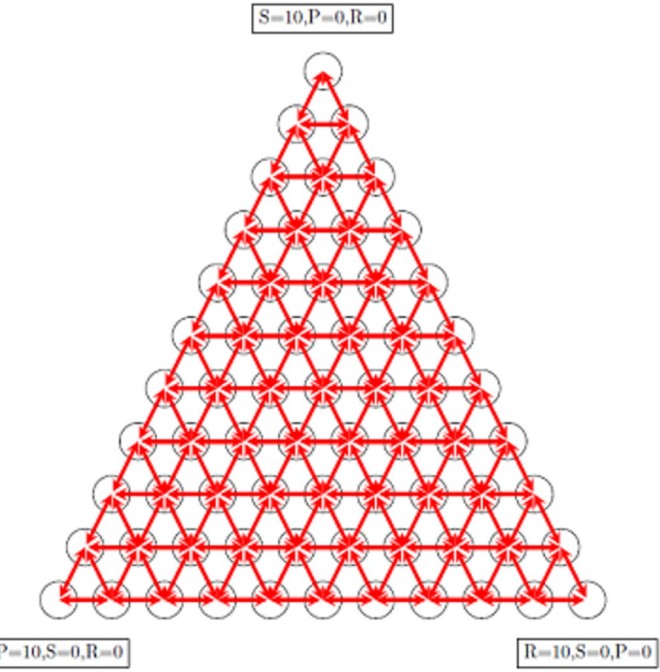

**Fig 8. The Markov chain corresponds to an ergodic RSP game for $N$ = 10.** The total number of states is 55 and the arrows determine the allowed transition between states. All arrows are two-way which means when the Markov chain is in a state there is a non-zero probability to escape from it.

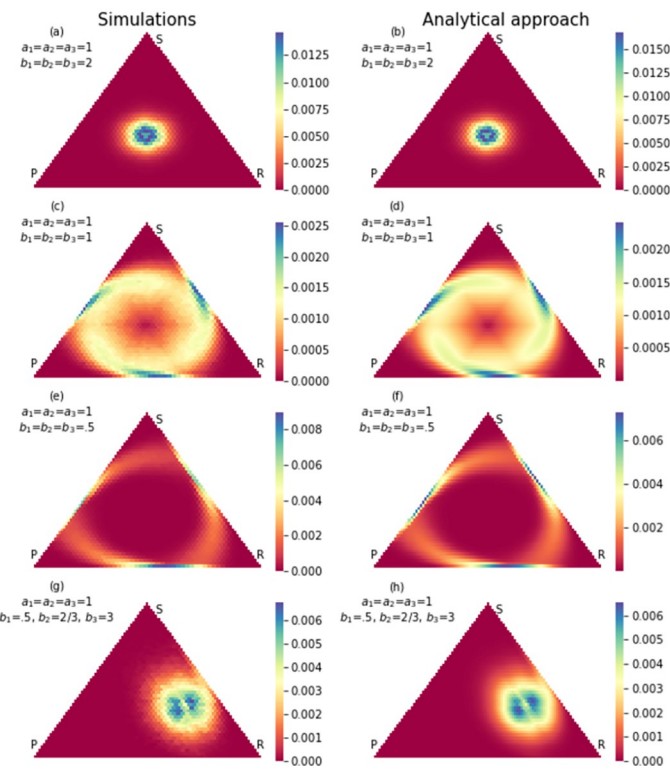

**Fig 9. Stationary probability distribution of RSP game with population 50.** The update rule is according to Eq 10. The payoff matrix in (a) and (b) is $a_i = 1$, $b_i = .5$ in (c) and (d) $a_i = 1$, $b_i = 1$ and in (e) and (f) $a_i = 1$, $b_i = 2$. To evaluate non-neutral selection in (g) and (h) the payoff matrix set as $a_i = 1$, $b_1 = .5$, $b_2 = 2/3$, $b_3 = 3$.

## Conclusion

This paper introduced the Markov chain method as an accurate analytical method for analyzing evolutionary game dynamics. Before this, the Makov chain method was used for studying two strategies evolutionary game or Moran process, but using the theorem explained in section, the Markov chain method can be used for any evolutionary game with any number of strategies. This method is flexible with changing the update rule of the evolutionary game. In the case of update rules which determine some fixation strategies, the fixation probability of each strategy and fixation time were calculable by the typical Markov chain method. By the theorem of section one can obtain conditional fixation time for each strategy. As an example, RSP games are evaluated with two update rules. In the first update rule, each of the three strategies can be fixed. Using the fundamental method, fixation probability and conditional fixation time of one of the strategies obtained were consistent with simulation results. In the second update rule, mutation is possible in the evolutionary game, and there is no fixed strategy. Getting the left null vector of matrix $P - I$ leads to the limited probability distribution in agreement with simulation results. This method could also be applied to evolutionary games with more than three strategies.

There is wide possibility of application of Markov chain method not only RPS game. In refrences [50, 51], we used Markov chain method for evaluating the Moran process. In many situations the issue of social dilemma represented by either Prisoner's Dilemma, Chicken, or Stag Hunt games [71, 72], therefore, applying this method on archetype $2 \times 2$ symmetric games will lead to significant results.

Codes related to this article is hosted on Github at https://github.com/mehdiphy/rock-scissors-paper-evolutionary-game.

## Appendix

In this appendix, we will prove the theorem of section

The theorem is about calculating conditional absorption time in absorbing the Markov chain. It has already been proven [53] that in absorbing the Markov chain the fundamental matrix $N = (I - Q)^{-1}$ is exists and can be written in an infinite series

$$N = I + Q + Q^2 + Q^3 + \tag{11}$$

Let $s_i$ and $s_j$ be two transient states. We assume that the chain starts in state $s_i$. Let $X^{(k)}$ be a random variable which equals 1 if the chain is in state $j$ after $k$ steps and equals 0 otherwise. Let $\mathcal{A}$ denote the outcome that corresponds to the absorbing of the chain to the absorbing state $s_a$. We need to calculate $P(X^{(k)} = 1|\mathcal{A})$ to obtain conditional absorbing time, $\tau_{ia}$. To this end, we use the following relation for conditional probability

$$P(X^{(k)} = 1|\mathcal{A}) = \frac{P(X^{(k)} = 1 \cap \mathcal{A})}{P(\mathcal{A})} \tag{12}$$

Clearly $P(\mathcal{A}) = \rho_i$ and $P(X^{(k)} = 1) = q_{ij}^{(k)}$.

Now, using

$$P(X^{(k)} = 1 \cap \mathcal{A}) = P(\mathcal{A}|X^{(k)} = 1)P(X^{(k)} = 1) = \rho_j q_{ij}^{(k)} \tag{13}$$

we arrive at

$$P(X^{(k)} = 1|\mathcal{A}) = \frac{q_{ij}^{(k)} \rho_j}{\rho_i} =: q_{ij}^{\prime(k)} \tag{14}$$

The expected number of times the chain is in state $s_j$ in the first $m$ steps given that it absorb in state $s_a$ and starts in state $si$ is

$$E(X^{(0)} + X^{(1)} + \ldots + X^{(m)}) = \frac{q_{ij}^{(0)} \rho_j}{\rho_i} + \frac{q_{ij}^{(1)} \rho_j}{\rho_i} + \ldots \frac{q_{ij}^{(m)} \rho_j}{\rho_i} \tag{15}$$

when $m$ goes to infinity we have

$$E(X^{(0)} + X^{(1)} + \ldots) = \frac{q_{ij}^{(0)} \rho_j}{\rho_i} + \frac{q_{ij}^{(1)} \rho_j}{\rho_i} + \ldots = n_{ij}^{\prime(k)} \tag{16}$$

Using these conditional probabilities we can calculate the conditional absorbing time, $\tau_{ia}$, as $\tau_{ia} = \sum_j n_{ij}'$ where $n_{ij}' = \sum_{k=0}^{\infty} q_{ij}^{\prime(k)}$.

This way, one can obtain the average conditional absorption time for the processes which are eventually absorbed to each arbitrary absorbing state.

## Acknowledgments

The authors would like to thank Arne Traulsen for reading the first draft of this paper and for useful comments and suggestions.

## Author Contributions

**Conceptualization:** Mahdi Hajihashemi, Keivan Aghababaei Samani.

**Data curation:** Mahdi Hajihashemi, Keivan Aghababaei Samani.

**Formal analysis:** Mahdi Hajihashemi, Keivan Aghababaei Samani.

**Investigation:** Mahdi Hajihashemi, Keivan Aghababaei Samani.

**Methodology:** Mahdi Hajihashemi, Keivan Aghababaei Samani.

**Software:** Mahdi Hajihashemi, Keivan Aghababaei Samani.

**Supervision:** Keivan Aghababaei Samani.

**Validation:** Mahdi Hajihashemi, Keivan Aghababaei Samani.

**Visualization:** Mahdi Hajihashemi, Keivan Aghababaei Samani.

**Writing – original draft:** Mahdi Hajihashemi, Keivan Aghababaei Samani.

**Writing – review & editing:** Mahdi Hajihashemi, Keivan Aghababaei Samani.

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
