## [Decision Letter · Decision Letter 0]

17 Jan 2022

PONE-D-21-40483Multi-strategy evolutionary games: a Markov chain approachPLOS ONE

Dear Dr. Hajihashemi,

Thank you for submitting your manuscript to PLOS ONE. After careful consideration, we feel that it has merit but does not fully meet PLOS ONE’s publication criteria as it currently stands. Therefore, we invite you to submit a revised version of the manuscript that addresses the points raised during the review process.

We look forward to receiving your revised manuscript.

Kind regards,

Jun Tanimoto

Academic Editor

PLOS ONE

Journal Requirements:

Reviewers' comments:

Reviewer's Responses to Questions

**Comments to the Author**

1. Is the manuscript technically sound, and do the data support the conclusions?

Reviewer #1: Yes

Reviewer #2: No

2. Has the statistical analysis been performed appropriately and rigorously? 

Reviewer #1: Yes

Reviewer #2: N/A

3. Have the authors made all data underlying the findings in their manuscript fully available?

Reviewer #1: Yes

Reviewer #2: No

4. Is the manuscript presented in an intelligible fashion and written in standard English?

Reviewer #1: Yes

Reviewer #2: No

5. Review Comments to the Author

Reviewer #1: This MS, supported by healthy science and mathematics, delivers a nice and interesting finding. Markov chain approach, although the authors said a ‘theoretical’ approach, provides the dynamics of a time-discrete system. In this sense, the methodology can be said an ‘approximation’ of an inherently time0continuous system. I would like to suggest the authors to note this remark somewhere to be appropriate in the revised MS.

After building up the formulation, the latter part shows numerical results applied to Rock-Paper-Scissors (RPS) game that is one of the architype symmetric games of 2-player & 3-strategy game class. The authors compared the theoretical result based on their approach with MAS result. Both are well consistent.

RPS game is a quite good template which may attract quite a few people dealing with EGT. In this sense, I suggest the authors to more deeply review some up-coming fruits on RPS game for recent years by citing some of them, e.g., The role of pairwise nonlinear evolutionary dynamics in the rock–paper–scissors game with noise, Applied Mathematics and Computation 394, 125767, 2021.

Besides RPS game, a question coming to mind is why the authors not trying to 2 by 2 game including Prisoner’s Dilemma (PD), Chicken, Stag Hunt (SH) and Trivial game classes that have been regarded as the most typical template in the filed of EGT. More importantly, the issue of social dilemma represented by either PD, Chicken, or SH might be more pressing in terms of application than RPS game. Although I wouldn’t go as far as to say that the authors should do with further series of Multi Agent Simulations (MASs), they should mention something somewhere (perhaps, at the end of conclusive remark) that there is wide possibility of application not only RPS game but also archetype 2 by 2 symmetric games by citing recent some review papers or books; e.g., (i) Sociophysics Approach to Epidemics, Springer, 2021, (ii) Evolutionary Games with Sociophysics: Analysis of Traffic Flow and Epidemics, Springer, 2019.

Reviewer #2: The authors studied the multi strategies (rock-paper-scissors) evolutionary game both in analytically and numerically using Markov chain model. Despite the interest and originality of the work, I could not fully evaluate its implementation and soundness. I have doubts about the novelty and the strength of advance required for the reputed journal. Therefore, at this stage, the manuscript cannot be recommended for publication.

6. PLOS authors have the option to publish the peer review history of their article (what does this mean?). If published, this will include your full peer review and any attached files.

Reviewer #1: No

Reviewer #2: No

---

## [Author Response · Author response to Decision Letter 0]

29 Jan 2022

We made the corrections suggested by respecred refrees.

---

## [Decision Letter · Decision Letter 1]

2 Feb 2022

Multi-strategy evolutionary games: a Markov chain approach

PONE-D-21-40483R1

Dear Dr. Hajihashemi,

We’re pleased to inform you that your manuscript has been judged scientifically suitable for publication and will be formally accepted for publication once it meets all outstanding technical requirements.

Kind regards,

Jun Tanimoto

Academic Editor

PLOS ONE

Additional Editor Comments (optional):

Reviewers' comments:

Reviewer's Responses to Questions

**Comments to the Author**

1. If the authors have adequately addressed your comments raised in a previous round of review and you feel that this manuscript is now acceptable for publication, you may indicate that here to bypass the “Comments to the Author” section, enter your conflict of interest statement in the “Confidential to Editor” section, and submit your "Accept" recommendation.

Reviewer #1: All comments have been addressed

Reviewer #2: All comments have been addressed

2. Is the manuscript technically sound, and do the data support the conclusions?

Reviewer #1: (No Response)

Reviewer #2: Yes

3. Has the statistical analysis been performed appropriately and rigorously? 

Reviewer #1: Yes

Reviewer #2: Yes

4. Have the authors made all data underlying the findings in their manuscript fully available?

Reviewer #1: Yes

Reviewer #2: (No Response)

5. Is the manuscript presented in an intelligible fashion and written in standard English?

Reviewer #1: Yes

Reviewer #2: Yes

6. Review Comments to the Author

Reviewer #1: The revised manuscript if this version by the authors seems adequately persuasive me. Thus, at this moment, I can suggest it as acceptable.

Reviewer #2: (No Response)

7. PLOS authors have the option to publish the peer review history of their article (what does this mean?). If published, this will include your full peer review and any attached files.

Reviewer #1: No

Reviewer #2: No

---

## [Editor Report · Acceptance letter]

8 Feb 2022

PONE-D-21-40483R1 

Multi-strategy evolutionary games: a Markov chain approach 

Dear Dr. Hajihashemi:

I'm pleased to inform you that your manuscript has been deemed suitable for publication in PLOS ONE. Congratulations! Your manuscript is now with our production department. 

Kind regards, 

on behalf of

Prof. Jun Tanimoto 

Academic Editor

PLOS ONE